# Daily Physical Activity in Asthma and the Effect of Mepolizumab Therapy

**DOI:** 10.3390/jpm12101692

**Published:** 2022-10-11

**Authors:** Marios Panagiotou, Nikolaos Koulouris, Antonia Koutsoukou, Nikoletta Rovina

**Affiliations:** 1st Department of Respiratory Medicine, Medical School, National and Kapodistrian University of Athens, “Sotiria” Chest Disease Hospital, 115 27 Athens, Greece

**Keywords:** asthma, severe asthma, daily physical activity, movement intensity, step count, moderate-to-vigorous physical activity (MVPA), accelerometer, guidelines, mepolizumab

## Abstract

For the various asthma-specific beneficial effects of physical activity, daily physical activity (DPA) and the potential of asthma therapies on DPA require better characterization. Hence, we aimed to determine (a) the DPA of asthma patients, and (b) the effect of add-on mepolizumab on the DPA of severe asthma patients. **Methods:** Adult outpatients with mild-to-moderate or severe asthma had accelerometer assessment of DPA. Severe asthma patients who were commenced on mepolizumab had their DPA reassessed after 12 months. **Results:** For the total cohort (*n* = 36), daily step count, time in moderate-to-vigorous physical activity (MVPA), MVPA volume and Movement Intensity (MI) were 7806 ± 3823 steps, 123 (interquartile range, 63) min, 657 ± 255 MET·min and 1.96 (0.45) m/s^2^, respectively. All patients met at least one recommendation for DPA but less than half met recommendations for vigorous DPA. Patients on mepolizumab therapy increased daily step count (646 steps; 9%), time in MVPA (20 min; 21%), MVPA volume (87 MET·min; 17%) and MI (0.11 m/s^2^; 6%) for the same amount of moving time; lung function, asthma control and health-related quality of life also improved. **Conclusions:** Analysis of the first national data on DPA in asthma and novel comparison against current applicable guidelines and identified beneficial thresholds showed borderline levels of DPA with room for improvement especially for severe asthma patients. In a non-sedentary cohort of severe asthma patients, mepolizumab conferred significant and meaningful improvements in DPA.

## 1. Introduction

Asthma is a heterogeneous chronic inflammatory disease of the airways that is defined by the history of respiratory symptoms such as wheeze, shortness of breath, chest tightness and cough that vary over time and in intensity, together with variable expiratory limitation [1]. With a prevalence of 1–18% in different countries, asthma is affecting millions of people across all ages and continues to carry substantial morbidity, mortality and socioeconomic burdens globally [1,2,3].

Following the modern paradigm of other chronic conditions, there is increasing focus on physical activity in asthma. Increasing insights into the nature of asthma and effective therapies have allowed for a paradigm shift towards active lifestyle. Enhancing and achieving normal levels of daily physical activity (DPA) is increasingly considered an important element for successful management and outcomes in asthma. Beyond benefiting all-cause mortality, risk of cancer and chronic disease, health-related quality of life (HRQoL) and other important aspects of general health, physical activity exerts asthma-specific effects. Consistent evidence links higher levels of physical activity with favorable pathophysiological and clinical effects in asthma including improved airway inflammation and hyperresponsiveness, lung function, asthma control, exacerbation rate, and healthcare use [4]. However, evidence also shows that patients with asthma, especially severe asthma, often engage in sedentary levels of physical activity and at lower levels than controls [5,6]. This line of evidence sets the scientific rationale for further study of the levels and patterns of physical activity in asthma but also, effective interventions to promote the physical activity of patients with asthma.

Biologic therapies are rapidly transforming the management and outcomes of patients with severe asthma. Although severe asthma has a low estimated prevalence of 3–10% in the total asthma population, it represents the most challenging and significant type of asthma by disproportionally affecting high on morbidity and mortality and accounting for a major share of healthcare utilization and cost as well as socioeconomic cost in asthma [1,7]. In the eosinophilic phenotype of severe asthma, mepolizumab, an anti-interleukin(IL)-5 biological therapy, reduces asthma exacerbation rates and corticosteroid use and improves asthma control and HRQoL [8,9,10]. However, the potential effect of mepolizumab -or other biological therapies- on DPA of patients with severe asthma is insufficiently explored.

Therefore, this study aims to determine both (a) the levels of DPA in patients with mild-to-moderate and severe asthma and cross-reference them with current guideline recommendations, and (b) whether mepolizumab add-on therapy improves DPA in patients with severe eosinophilic asthma.

## 2. Materials and Methods

### 2.1. Patients

The study was conducted in a prospective, observational design. It consisted of two arms of parallel design: (1) the physical activity arm, and (2) the mepolizumab arm. For the physical activity arm, consecutive adult patients with adequately established and documented asthma of any severity who attended the Asthma Outpatient Clinic, “Sotiria” Chest Diseases Hospital were eligible. Eligible patients for the mepolizumab arm were those who additionally met the definition of severe asthma [7] and predetermined criteria for mepolizumab therapy [11]. The decision-making for add-on mepolizumab therapy in eligible patients was based exclusively on clinical criteria. Patients were administered 100 mg of mepolizumap (Nucala; London, GlaxoSmithKline, UK) once every four weeks for 12 months by subcutaneous injection into the upper arm. Exclusion criteria for both study arms included current or recent (≤1 month) severe exacerbation [7] and comorbidities interfering with physical activity such as musculoskeletal and neurological conditions.

### 2.2. Clinical Assessment

At the study entry point, the most recent values of forced expiratory volume in 1 s (FEV_1_), forced vital capacity (FVC) and eosinophil blood count were retrieved from the medical record. All subjects completed the Asthma Control Test (ACT) and the “past 4 weeks” version of St. George’s Respiratory Questionnaire (SGRQ). Patients on the mepolizumab arm also completed the Global Rating of Activity Limitation questionnaire (GRALQ) before and after therapy, and the Global Impression of Change in Activity Limitation questionnaire (GICALQ) after therapy.

### 2.3. Monitoring of Daily Physical Activity

DPA was captured using a commercially available triaxial trunk accelerometer and read with a specific software (DynaPort MoveMonitor and DynaPort Manager, respectively; McRoberts B.V., The Hague, The Netherlands). The MoveMonitor device has a sample frequency of 100 samples/s and a range of −6 g to +6 g and has been extensively validated and used for activity detection and energy expenditure estimation in younger as well as older healthy populations and in chronic obstructive pulmonary disease (COPD), in both ordinary and sedentary conditions [12,13,14,15,16,17,18]. The accelerometers were attached to an elastic strap and mounted on the subjects’ back, at the level of the L2 vertebra to wear at all times for seven consecutive days, excluding aquatic activities since this would damage the device. Data from days on which the accelerometer was worn more than 75% of the time (≥18 h), were averaged to determine habitual DPA; a minimum of 5 valid wearing days was required for a participant’s data to be included in the analysis [19].

DPA was assessed through measurement of daily step count, daily time spent in moderate-to-vigorous physical activity (MVPA; ≥3 METs) (min), daily MVPA volume (METs.min), daily Movement Intensity (MI; m/s^2^; 9.81 m/s^2^ = 1 unit of gravitational acceleration g) and daily moving time (min), as provided by the algorithm for all whole-body moving activities (including walking, stair walking and cycling). We calculated the daily time spent in MVPA by summing the time in moderate physical activity (3–5.9 METs) and the time in vigorous physical activity (≥6 METs). In order to compare against guidelines for physical activity [20], we calculated the weekly time in moderate activity, weekly time in vigorous activity and weekly MET·min count from the daily data. Bouts of any duration were considered for all the activities [20].

### 2.4. Physical Activity Guidelines

Updated guidelines by the World Health Organization (WHO) advise that adults should engage in at least 150–300 min of moderate-intensity (3–5.9 metabolic equivalents; METs) aerobic physical activity throughout the week or do at least 75–150 min of vigorous-intensity (≥6 METs) aerobic physical activity throughout the week or an equivalent combination of moderate-and vigorous-intensity activity (MVPA; ≥3 METs), preferably spread evenly over four to five days a week, or every day [20]. When combining moderate and vigorous intensity activity to meet the current recommendation, the minimum goal should be in the range of 600 to 1200 MET·min per week [20,21]. In light of new evidence [22,23,24], the updated guidelines take into consideration the activity of any bout duration [20]. These recommendations are relevant to all healthy adults aged 18 and older (including aged 65 and older), irrespective of gender, race, cultural background or socioeconomic status, and are relevant to people of all abilities including people with chronic non-communicable medical conditions not related to mobility and thus applicable to patients with asthma.

### 2.5. Statistical Analysis

Data were analyzed using the SPSS statistical package v28.0. Normal distribution of the data was checked using the Shapiro-Wilk test. Data are expressed as mean ± standard deviation or median (interquartile range) for continuous outcomes, and as counts and percentages for categorical variables. Independent samples t-tests or Mann–Whitney U-tests were used for comparisons of differences in clinical characteristics and DPA parameters (daily step count, time in MVPA, MVPA volume, MI and moving time) between patients with mild-to-moderate asthma and patients with severe asthma. Chi-square tests were used for detection of between-group differences in compliance with guideline recommendations. Associations of DPA parameters with FEV_1_, ACT score and SGRQ score as well as associations between all of the DPA parameters were examined using Pearson’s or Spearman’s correlation coefficient, depending on the distribution of the data. Paired samples t-tests or Wilcoxon signed rank tests were used to examine for changes with therapy in DPA parameters, FEV_1_, ACT score, SGRQ score and eosinophil count. The level of significance was set at *p* < 0.05.

## 3. Results

### 3.1. Study Flow

A total of 38 patients were enrolled between January 2018 and March 2020. The study was prematurely concluded due to the advent and persistence of the coronavirus disease 2019 (COVID-19) pandemic in Greece. The decision was made for health and safety reasons but also due to the prolonged public mobility restrictions and lockdowns imposed at a national and global level, which could introduce systematic error (bias). Two patients (6%) with mild-to-moderate asthma were excluded due to poor wearing compliance. Thirty-six patients were therefore considered in the analysis for the physical activity arm of the study, of whom 15 had mild-to-moderate asthma and 21 had severe asthma. Sixteen of the patients with severe asthma had eosinophilic asthma and commenced on mepolizumab therapy. Twelve patients successfully completed 12-month therapy and were included in the analysis for the mepolizumab study arm; two patients dropped out due to the worsening of asthma symptoms, and 2 patients were excluded for not following the instructed medication (Figure 1). Of note, sporadic, paradoxical worsening of asthma symptoms and lung function with mepolizumab therapy has been reported [25,26].

### 3.2. Patient Characteristics

The complete patient characteristics are shown in Table 1. There was no difference in sex, age or body mass index (BMI) between patients with mild-to-moderate asthma and severe asthma, but patients with severe asthma had lower FEV_1_ (*p* = 0.001), ACT score (*p* < 0.001) and SGRQ score (*p* < 0.001). All of the patients were on regular therapy with at least a combination of inhaled corticosteroids and long-acting beta agonists.

### 3.3. Daily Physical Activity

For the whole patient cohort, the daily step count was 7806 ± 3823 steps; time in MVPA 123 (interquartile range, 63) min, MVPA volume 657 ± 255 MET·min, MI 1.96 (0.45) m/s^2^ and moving time 94 ± 43 min. As shown in Table 2, despite considerable mean differences (MD) in the above DPA parameters in favor of patients with mild-to-moderate asthma compared to patients with severe asthma, the between-group differences were not statistically significant.

For the whole patient cohort, there were significant associations between DPA and FEV_1_ (r for step count = 0.40, *p* = 0.018; r for time in MVPA time = 0.36, *p* = 0.036; r for MVPA volume = 0.42, *p* = 0.012; r for MI = 0.44, *p* = 0.008). Daily MI (but not any of the other DPA parameters) associated with ACT score (r= 0.37, *p* = 0.027). There were no associations between any DPA parameter and SGRQ score (r= 0.30–0.33, *p* = 0.052–0.090 for all). There were strong associations between daily step count, time in MVPA, MVPA volume and MI and moving time (r = 0.50–0.96, *p* = 0.001–0.003 for all).

### 3.4. Compliance with Physical Activity Guidelines

The percentage of all patients who met current recommendations for moderate, vigorous or combined moderate and vigorous daily physical activity was 100%, 42% and 100%, respectively, without between-group differences (Table 2).

### 3.5. The Effect of Mepolizumab Therapy

After 12-month therapy, the 12 patients (10 female; age 57.4 ± 8.7 years; BMI 29.8 ± 3.5 Kg/m^2^, FEV_1_% pred. 68.6 ± 26.5; ACT score 15 (5); SGRQ score 58.8 (34.5) on the mepolizumab arm showed significant increase in daily step count (from 7233 ± 3994 to 7879 ± 3892 steps, *p* = 0.009), daily time in MVPA (from 97(70) to 117(64) min, *p* = 0.003), daily MVPA volume (from 519 ± 164 to 606 ± 269 MET·min, *p* < 0.001) and daily MI (from 1.72 (0.50) to 1.83 (0.48) m/s^2^, *p* = 0.003) (Figure 2). There was no change in daily moving time before and after therapy (87 ± 43 versus 89 ± 34 min; *p* = 0.906).

Patients also exhibited meaningful improvement in FEV_1_ (from 1.74 ± 0.8 to 1.88 ± 0.8, *p* = 0.004), ACT score (from 15.73 ± 4. to 21.1 ± 4.8, *p* = 0.003) [27] and SGRQ score (from 52.3 ± 20.7 to 30.6 ± 21.9, *p* = 0.004) [28]. There was significant change in the self-reported GRALQ with all the patients rating their physical activity as “much better (25%)”, “better (50%)” or “slightly better (25%)” than prior to therapy. In GICALQ, the proportion of patients rating their activity as “not limited” or “slightly limited” increased from 25% to 75% (Figure 3). As expected, after 12 months of mepolizumab therapy, the eosinophil count was reduced (from to 5.6 ± 2.6 to 1.2 ± 0.6, *p* < 0.001), but not eliminated.

## 4. Discussion

In this study of patients with moderate-to-severe and severe asthma, the levels of DPA are deemed borderline satisfactory. All the patients met at least one recommendation for DPA but less than half met recommendations for vigorous DPA and there was space for improvement in all the aspects of DPA, especially for patients with severe asthma. Importantly, in a non-sedentary cohort of older patients with severe asthma, mepolizumab increased meaningfully daily step count by 646 steps (9%), time in MVPA by 20 min (20%), MVPA volume by 87 MET·min (17%) and MI by 0.11 m/s^2^ (6%). This accelerometer-derived improvement matched self-reported improvement in DPA. Clinically significant improvements also occurred for lung function, asthma control and HRQoL.

Individually, all patients met at least one of the recommended thresholds for moderate activity, vigorous activity or an equivalent combination of MVPA [20]. However, the absolute compliance with recommendations was driven by performance in moderate-intensity physical activity rather than in vigorous physical activity (VPA). VPA compliance for all patients, mild-to-moderate asthmatics and severe asthmatics was 41%, 53% and 38%, respectively with some individuals showing negligible or absent VPA. This is important since VPA is an essential aspect of DPA. In a national cohort study of 403,681 adults, VPA showed a stronger inverse association with cancer mortality compared with moderate physical activity [29]. Also, among participants performing any MVPA, a higher proportion of VPA to total physical activity was associated with lower all-cause mortality across sociodemographic characteristics, lifestyle risk factors, and chronic conditions at baseline [29].

In terms of the weekly MVPA volume, the performance of our patients (3828 ± 2651 MET·min/week for severe asthmatics and 4323 ± 1707 MET·min/week for mild-to-moderate asthmatics) may seem in excess in relation to the recommended range of 600 to 1200 MET·min/week [20,21]. However, recommendations refer merely to the minimal required amount of activity in order to achieve substantial benefits over and above the routine light-intensity activities of daily living. Strong evidence suggests that total physical activity needs to be several times higher (≥3000–4000 MET·min/week), than the recommended minimum level to achieve significant health benefits such as reducing the risk of breast cancer, colon cancer, diabetes, ischemic heart disease, and ischemic stroke events [30].

The daily step count was 7806 ± 3823 steps for the total cohort, 8209 ± 3815 steps for patients with mild-to-moderate asthma, and 7518 ± 3896 steps for patients with severe asthma. There are still no evidence-based public health guidelines recommending a range for steps per day for health benefits to cross-reference our data against. However, the patients with severe asthma only reached the threshold of the “somewhat active” category (7500–9999 steps per day) in a widely endorsed step-defined ladder of physical activity [31,32]. Crucially, many patients did not achieve full potential of benefit from the daily step count. Robust evidence shows that an increasing daily step count is associated with progressively lower mortality risk, with the risk plateauing for older adults (aged ≥ 60 years) at approximately 6000–8000 steps per day and for younger adults (aged < 60 years) at approximately 8000–10,000 steps per day [33], but only 55% of our study population walked within these ranges.

The observed increase in DPA with mepolizumab therapy is meaningful as it translates directly to clinical benefits and helps the patients to achieve beneficial goals for physical activity. Strong evidence shows that increments in daily step count as well as duration or frequency of MVPA are associated with a progressively lower risk of all-cause mortality [23,34,35,36], and current recommendations are that additional physical activity of any bout duration and intensity, accumulated over the day and week, is beneficial [20,37]. Importantly, the achieved mean increase of 646 steps per day with mepolizumab therapy matched the minimal important difference for hospital admission in COPD patients after 3-month multidisciplinary pulmonary rehabilitation, estimated to be 600–1000 steps per day [38]. It also surpassed the threshold of 500-step increment of steps per day shown to lower the risk of cardiovascular disease by 6%, and approached the 1000-step increment of steps per day that was associated with a 23% decreased risk of all-cause mortality [36]. Interestingly, there was no concurrent change in daily moving time with mepolizumab, which means that patients achieved higher levels of DPA via increasing the intensity rather than the duration of their activities.

To our knowledge, the present is the second study to investigate the effect of any asthma therapy on DPA. Carpagnano et al. [39] found significant improvements in daily step count and energy expenditure in 30 patients treated with biological therapy (omalizumab or mepolizumab) for six months compared to patients on traditional therapy. The magnitude of improvement with biologic therapy was notably larger than in our study, as the patients almost doubled their daily steps (from 3806 ± 421 to 6545 ± 844) [39]. However, those patients were highly sedentary compared to our patient cohort, which certainly allowed a larger space for improvement.

Herein, we provide the first data on MI in asthma expressed in acceleration units, being 1.96 (0.45) m/s^2^ for the total cohort. The MI in m/s^2^ represents a direct measure of the absolute power of movement. It has been extensively investigated in COPD, where it was shown to be an important aspect of DPA, ranging 1.5 to 1.9 m/s^2^ depending on the disease severity and study population [40,41,42] and responsive to pharmacological and exercise interventions [43]. With a mean MI of 1.72 (0.50) m/s^2^, our patients with severe asthma showed similar performance to generally older patients with at least moderate COPD. Finally, we observed modest associations of DPA with FEV_1_. This is in line with the finding by Hennegrave et al. [44] that FEV_1_ associated modestly with daily step count in asthmatic patients. van’t Hul et al. [45] and Bahmer at al. [46] found no associations between accelerometer-derived parameters of DPA and spirometric measures or peak expiratory flow rates in asthma, but daily step count was associated with impulse oscillometric airway resistance and small airway dysfunction [46].

A review of previous available studies on DPA in severe asthma patients is provided in Table 3. Notably, there is considerable variation in the levels of DPA among the studies, ranging from low sedentary to acceptably active. Possible causes include variations in cultural and behavioral patterns, seasonality as well as disparities in methodology and the way that data are presented. The type of the device, wear location, speed and duration of movement can all affect the accuracy of measures obtained from different devices; counts obtained from different devices are highly correlated but can vary significantly [33]. Compared to four studies that used an arm-positioned accelerometer and two studies that used an above hip-positioned accelerometer, we used an accelerometer that is positioned on the back, at the level of the L2 vertebra, an approximation of the body’s center of mass.

The main limitation of this study is the small sample size, which reduces the generalizability of results. As suggested by the large mean differences, significant between-group differences in DPA as well as significant associations of DPA may have been overlooked due to the inadequate statistical power. Another limitation is the absence of a control group for the mepolizumab arm to strengthen the results of the study. Our data is from a single center, but our patients are considered representative of a public referral center for asthma and severe asthma in Greece. Finally, the authors would like to acknowledge the negative impact of the COVID-19 pandemic, leading to the premature termination of the study as the public physical activity remains limited compared to the pre-pandemic era. Ongoing recommendations for physical distancing, quarantine measures for people with COVID-19, individual precautionary behaviors and fear of disease are leading to the adaptation of new habits that continue to bear a strong negative impact on day-to-day life physical activity and sedentary behavior worldwide [49,50]. This is an unfortunate, important consideration for any current or future study in physical activity.

## 5. Conclusions

By showing meaningful improvements in several parameters of the DPA of patients with severe asthma with mepolizumab therapy, this study adds to the limited available evidence on the effect of biologic therapy on physical activity in asthma. Larger multicenter studies are needed to validate our preliminary results, but the potential of a single, add-on intervention on improving such an important patient-centered and challenging outcome is appealing. For the multiple significant general health- and asthma specific- benefits of physical activity, DPA reflects long-established behavioral patterns that are often difficult to amend even when the disease severity and control improve. Longer-lasting interdisciplinary approaches targeting beyond asthma-specific endpoints may therefore be required to help patients meet recommendations and achieve their maximum potential in DPA and lifestyle pursuits. This is an important consideration not only for patients with severe asthma who plausibly draw the major share of interest, but for all the patients with asthma, who often present suboptimal levels of DPA.

## Figures and Tables

**Figure 1 jpm-12-01692-f001:**
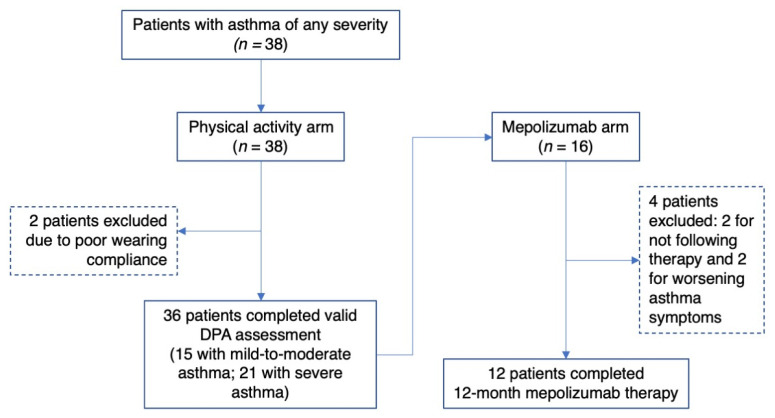
Study flowchart. DPA: Daily physical activity.

**Figure 2 jpm-12-01692-f002:**
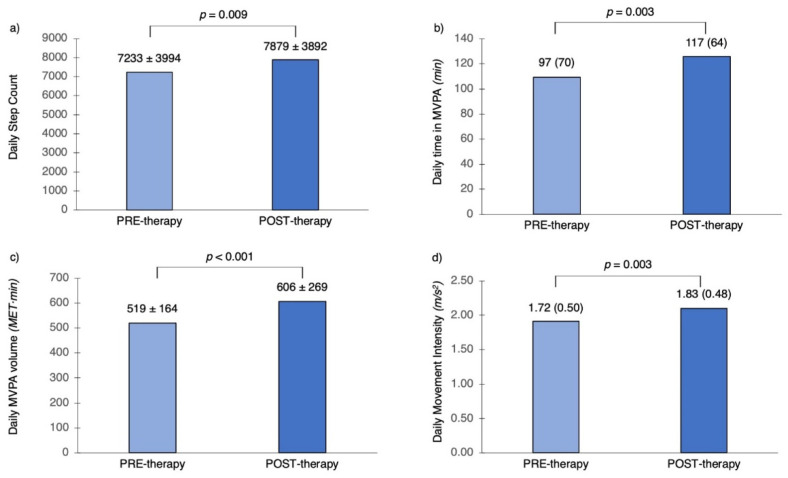
Changes in daily physical activity with mepolizumab therapy. Data are presented as the mean ± SD or median (interquartile range). MVPA: moderate-to-vigorous physical activity; MET: metabolic equivalent.

**Figure 3 jpm-12-01692-f003:**
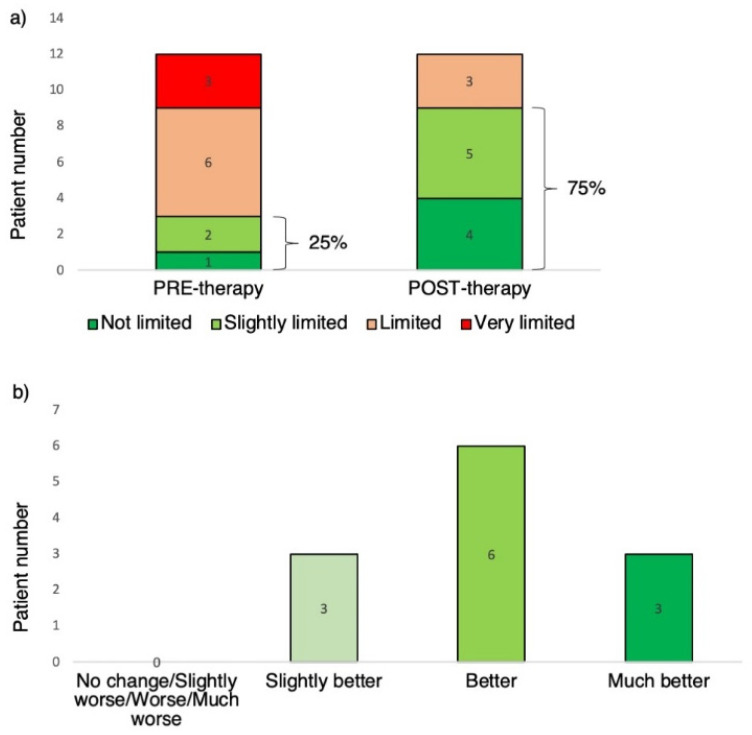
(**a**) Global Rating of Activity Limitation before and after mepolizumab therapy; (**b**) Global Impression of Change in Activity Limitation after mepolizumab therapy.

**Table 1 jpm-12-01692-t001:** Patient characteristics.

	All (*n* = 36)	Mild-to-Moderate Asthma (*n* = 15)	Severe Asthma (*n* = 21)	*p*
Sex, F/M	29/7	11/4	18/3	0.362
Age	49.8 ± 14.8	45.1 ± 16.5	53.1 ± 12.8	0.172
BMI (Kg/m^2^)	29.3 ± 6.8	27.7 ± 4.6	30.3 ± 7.9	0.238
FEV_1_ (L)	2.4 ± 1.1	3.1 ± 1.17	1.9 ± 0.8	<0.001
FEV_1_% predicted	83.4 ± 26.4	96.1 ± 19.2	73.9 ± 27.4	0.006
FEV_1_/FVC ratio	0.74 ± 1.5	0.79 ± 0.1	0.70 ± 0.2	0.084
Therapy				
ICS/LABA	36	15	21	N/A
LAMA	20	1	19	N/A
Montelukast	22	7	15	N/A
OCS	2	0	2	N/A
Omalizumab	2	0	2	N/A
ACT score	20 (9)	25 (4)	15 (5)	<0.001
SGRQ score	30.8 (44.1)	58.8 (34.5)	12.6 (10.8)	<0.001

Data are presented as the mean ± SD, median (interquartile range) or counts. FEV_1_: prebronchodilator forced expiratory volume in 1 second; FVC: prebronchodilator forced vital capacity; ICS: inhaled corticosteroids, OCS: oral corticosteroids; LABA: long-acting beta agonist; LAMA: long-acting muscarinic agonist; ACT: Activity Control Test; SGRQ: St George Respiratory Questionnaire; N/A, not applicable or not assessed.

**Table 2 jpm-12-01692-t002:** Daily physical activity and comparison against current guidelines.

	All (*n* = 36)	Mild-to-Moderate Asthma (*n* = 15)	Severe Asthma (*n* = 21)	*p*
**Daily step count**	7806 ± 3823	8209 ± 3815	7518 ± 3896	0.558
**Daily time in MVPA** (min)	123 (63)	133(64)	97 (70)	0.505
**Daily MVPA volume** (MET·min)	657 ± 255	618 ± 244	547 ± 266	0.553
**Daily Movement Intensity** (m/s^2^)	1.96 (0.45)	2.09(0.46)	1.72 (0.50)	0.083
**Daily moving time** (min)	94 ± 43	98 ± 44	91 ± 43	0.630
**Patients met recommendations for weekly time in moderate activity** [min/week] [20]	36; 100% [745 (392)]	15; 100% [742 (357)]	21; 100% [777 (389)]	1
**Patients met recommendations for weekly time in vigorous activity** [min/week] [20]	15; 41% [70 (137)]	8; 53% [126 (168)]	8; 38% [28 (112)]	0.464
**Patients met recommendations** **for weekly MVPA volume** [MET·min] [20,21]	36; 100% [4034 ± 1791]	15; 100% [4323 ± 1707]	21; 100 [3828 ± 2651]	1

Data are presented as the mean ± SD, median (interquartile range) or counts and percentages. MVPA: moderate-to-vigorous physical activity; MET: metabolic equivalent.

**Table 3 jpm-12-01692-t003:** Studies on daily physical activity in severe asthma.

Study	Sample Size	Accelerometer (Position)	Daily Step Count	Daily Time in MVPA	Daily Step Count Associations
Bahmer et al. [46]	63	SenseWear Pro Armband (upper arm)	6174 (4822–9277)	125 (68–172)	Impulse oscillometric airway resistance and small airway dysfunction
Cordova-Rivera et al. [5]	61	ActiGraph wGT3X-BT (over dominant hip)	5362 (3999–7817)	22 (13–38)	6MWD, ACT score, FEV_1_, hs-CRP
Cordova-Rivera et al. [47]	62	ActiGraph wGT3X-BT (over dominant hip)	5385 (3941–7844)	22 (13–35)	6MWD, FEV_1_% pred., dyspnoea, hs-CRP, eosinophils%
Carpagnano et al. [39]	40	SenseWear Pro Armband (upper arm)	3806 ± 421	N/A	
Hennegrave et al. [44]	23	SenseWear Pro Armband (upper arm)	6560 ± 3915	120 ± 54	Age, anxiety, FEV_1_
Neale et al. [48]	48	SenseWear Pro3 Armband (upper arm)	5183 ± 2935	44 ± 46	EQ-5D-3L, AQLQ score, CRQ score

Data are presented as Mean ± SD or median (interquartile range). MVPA: moderate-to-vigorous physical activity; 6MWD: 6-min walk distance; ACT: Asthma Control Test; FEV_1_: forced expiratory volume in 1 second; hs-CRP: high-sensitivity C-reactive protein; AQLQ: Asthma Quality of Life Questionnaire; CRDQ: Chronic Respiratory Disease Questionnaire.

## Data Availability

The data presented in this study are available in this article.

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
