# Peer review of "Daily Physical Activity in Asthma and the Effect of Mepolizumab Therapy"

_jpm, 2022, doi:10.3390/jpm12101692_

Round 1

Reviewer 1 Report

In this study, the authors determined the levels of daily physical activity (DPA) in asthmatic patients and the effects of mepolizumab add-on therapy in patients with severe eosinophilic asthma. They found that not all patients met recommendations for vigorous DPA and that DPA increased with mepolizumab therapy. The topic is interesting as there is a great interest to assess the level of physical activity in asthmatic patients. However, the results of this study may not be definitively validated mainly, because of the small sample size and other studies are then required to validate these preliminary results. Moreover, this manuscript may be improved mainly in its methodology and discussion parts.

Below are some specific comments:

Introduction:

The problematic was clearly described and the aims of this study were correctly defined and justified.

Materials and methods

- Page 2; line 66 to line 78: the study protocol is not well described: the authors, described only the recruitment of patients without describing the next steps. It is important to report (in a figure if possible) the study flow chart with describing the two steps of this study beginning from the initial assessment of asthmatic patients and then, the second evaluation following 12 months of mepolizumab therapy of severe asthmatic patients.

- Page 2; line 69: what about the age and sex of patients included in this study (only adults?, …)

- Page 2; line 77: some examples of comorbidities interfering with physical activity should be mentioned.

- Page 2; line 81: “Forced vital capacity” is more appropriate to use than “forced volume capacity”

- Page 3; line 125: part statistical analysis should more detailed or alternatively preceded by a “data analysis” part: it is important to describe the variables (from clinical assessment and daily physical activity) that were evaluated or compared at the two steps of this study and the comparisons that were made before describing the statistical tests used.

Results:

The different paragraphs in this part need to be simplified and should include only a description of data in tables and figures: it is not usefulness to repeat in text numbers or values that are mentioned in tables and figures.

Moreover, all the values should be checked (for example, in page 4; line 168: the MVPA volume did not correspond to the value of the daily MVPA volume mentioned in table 2)

Discussion

There is a lack of coherence in part discussion:

1- After reporting the main results of the study (page 7; line 216 to line 219), a discussion of methodology is lacked before discussion of results.

2- Discussion of the results should begin by the first finding of this study about daily physical activity in asthmatic patients. Nextly, discussion of the effects of mepolizumab therapy should follow.

Reviewer 2 Report

The authors have correctly pointed out the main limitation of this research being a single centre study with a very small sample size of 36 patients with only 12 patients in the treatment arm, making it quite difficult to derive any solid conclusions for the study. 

Some revisions required: 

The studied DPA parameters need to be better explained in the methodology section as there are inconsistencies and may sound very confusing to the general reader. These parameters can then be presented in more detail under the results section as it currently sounds very confusing - the authors have used words such as "modest significant associations, correlated, trend for modest associations" which does not add any value and can be misleading. 

The discussion is the main limitation of the study. Proper paragraphing is required with only one main idea in each paragraph followed by supporting evidences and correlations with other research. There is also a lack of flow with the authors jumping from one discussion idea to another. 

Another major limitation was is there a placebo arm to the treatment (mepolizumab arm) which would further strengthen the results of the study. 

Some minor grammatical errors will need to be looked at such as the opening statement of the abstract. 

Round 2

Reviewer 1 Report

The authors improved the quality of the manuscript. However, some concerns remain. Below are the comments:

Materials and methods:

- Page 3 ; line 134 : the groups that were compared were not defined (a group of patients with mild to moderate asthma and a group with severe asthma?)

- Page 3 ; line 140 to 143 : these lines should appear just before line 136 to be coherent with the order of data presented in part results.

Results:

- Page 5 ; line 182 to line 184 : The values appear in table 2 ; then, there is no need to repeat them in the text.

- Page 6; line 214 to 218: The values appear in figure 2 ; then, there is no need to repeat them in the text. It is sufficient to indicate only the type of change (an increase) of the different variables studied after mepolizumab therapy and its significance.

Discussion:

- Page 8; line 239: “patients with mild to moderate” instead of “patients with moderate to severe”

- Page 8; line 242: It seems that the improvement after mepolizumab therapy (which should be mentioned) concerned “almost all” aspects of DPA and not all aspects of DPA: the authors did not found after treatment a significant change in daily moving time which can be considered also an aspect of DPA.

- Globally, the authors improved the rearrangement of paragraphs in part Discussion and exposed finally a methodological discussion where they indicated some of the limitations of the methodological aspects in this study.

Author Response

Once again, we would like to extend again our appreciation for your insightful and helpful comments which we used to revise our manuscript.